# Multifunctionality of the Telomere-Capping Shelterin Complex Explained by Variations in Its Protein Composition

**DOI:** 10.3390/cells10071753

**Published:** 2021-07-11

**Authors:** Claire Ghilain, Eric Gilson, Marie-Josèphe Giraud-Panis

**Affiliations:** 1Université Côte d’Azur, CNRS, INSERM, IRCAN, 06000 Nice, France; claire.ghilain@unice.fr; 2International Research Laboratory for Cancer, Aging and Hematology, Shanghai Ruijin Hospital, Shanghai Jiaotong University and Côte-d’Azur University, Shanghai 200025, China; 3Department of Genetics, CHU Nice, 06000 Nice, France

**Keywords:** telomere, aging, Shelterin, senescence, DNA damage response

## Abstract

Protecting telomere from the DNA damage response is essential to avoid the entry into cellular senescence and organismal aging. The progressive telomere DNA shortening in dividing somatic cells, programmed during development, leads to critically short telomeres that trigger replicative senescence and thereby contribute to aging. In several organisms, including mammals, telomeres are protected by a protein complex named Shelterin that counteract at various levels the DNA damage response at chromosome ends through the specific function of each of its subunits. The changes in Shelterin structure and function during development and aging is thus an intense area of research. Here, we review our knowledge on the existence of several Shelterin subcomplexes and the functional independence between them. This leads us to discuss the possibility that the multifunctionality of the Shelterin complex is determined by the formation of different subcomplexes whose composition may change during aging.

## 1. Foreword

The linear nature of eukaryotic chromosomes causes two serious threats to genome integrity. The first threat stems from DNA extremities, which can be misidentified as DNA damage by the DNA damage response (DDR) machinery, leading to cellular senescence, apoptosis or double-strand break (DSB) repair [1]. The second threat stems from the inability of the conventional replication machinery to fully replicate the extremities of parental DNA. In somatic cells, this leads to an inexorable erosion of chromosomes ends, which is compensated for by activating pathways that replenish telomeres (Telomerase or Alternative Lengthening of Telomeres (ALT), based on recombination) in germ, stem and cancer cells [2]. In addition to this replicative attrition, telomeres are sensitive to a wide range of endogenous and environmental factors, such as improper cell cycle progression through mitosis, oxidative or genotoxic stress, alcohol, caffeine, heat shock, stress hormones and psychological stress [3]; therefore, telomeres have emerged as important cell-cycle and senescence regulators, stress sensors, and lifespan predictors. Indeed, experimental and pathological evidence of genome-wide deleterious consequences of telomeres dysfunction are numerous. For example, several premature aging syndromes generically called telomeropathies originate from or are associated with mutations in telomere-associated proteins (*Dyskeratosis congenita*, Hoyeraal–Hreidarsson, Revesz or Coats plus syndromes, amongst others) [4]. Furthermore, replenishing telomeres is an obligatory step for oncogenesis. Failure to do so causes rampant genome instability (multiple genome rearrangements through cycles of break–fusion–breaks, kataegis or chromothripsis) and cell death [5]; thus, the obvious importance of telomeric actors has elicited various therapeutic trials. Although Telomerase could be perceived as a prime target, limitations in Telomerase-based strategies, such as the consequential activation of ALT-driven telomere elongation or the possible pro-aging side effects, have stimulated the development of alternative approaches [6]. Targeting the complexes that form telomeres themselves could, thus, be a complementary or alternative route for telomere-based therapies.

We are soon reaching the 30 years benchmark since the discovery of the first mammalian telomeric protein, TRF1 [7,8,9]. Through the years, 5 other proteins (TRF2, RAP1, TIN2, TPP1 and POT1) have also been identified as being essential for the protection of these natural DNA ends and have been proposed to form a whole complex named Shelterin by de Lange in 2005 [10] (Figure 1).

## 2. The Shelterin Complex: Composition and Biological Role

The Shelterin ultimate role is to maintain telomeres homeostasis, which as we have seen, is critical for genome stability and cell fate. Moreover, telomeric proteins participate in several cellular processes not directly related to telomere homeostasis, such as replication, mitosis, meiosis, heterochromatin stability, immunity or neuronal development [11]; hence, the Shelterin complex is highly multifunctional and plays pivotal roles in telomere protection, genome stability and cell fate. Remarkably, each Shelterin subunit has specific functions in these processes, leading us to question the respective contributions of the whole complex versus subcomplexes and whether the protein composition of the Shelterin complex changes during aging.

On a molecular level, this complex is vital for efficient replication of the G-rich and repetitive telomeric DNA [12,13,14] and to regulate Telomerase- or ALT-driven elongation of telomeres. Importantly, the six telomeric proteins protect chromosome ends from the DNA damage response (through inhibition of the ataxia–telangiectasia mutated kinase (ATM) and the ataxia–telangiectasia and Rad3-related kinase (ATR) pathways) and from repair machineries (non-homologous end joining (NHEJ) and homologous recombination (HR)) [15].

Biochemical and structural studies have revealed interactions between sets of these proteins that have led to the definition of the complex containing the 6 proteins. The concept of Shelterin was not built in one day. It took more than 10 years from the identification of TRF1 by de Lange’s laboratory in the early 1990s [8] until the publication of TPP1 in 2004 simultaneously by three different groups [16]; however, deciphering the role of each of the members took decades, and even to this day new connections and functions of these proteins are being discovered. One crucial notion concerning the Shelterin proteins is their roles as hubs: recruiting proteins, regulating activities and controlling conformation of many different molecules, the first one being telomeric DNA. Binding telomeric DNA is the action of TRF1, TRF2 and POT1.

## 3. The DNA Binders: POT1, TRF1 and TRF2

Human POT1 (or Pot1a and Pot1b, the mouse equivalents of hPOT1) binds the single-stranded G-rich overhang that constitutes the ends of most telomeres throughout the eukaryotic kingdom and that serves as a substrate for Telomerase. This binding is mediated by two structural domains, called OB-folds (Oligonucleotide–Oligosaccharide binding fold [17], Figure 2) located at the N-terminus of the protein. A third domain, which contains a Holliday junction resolvase-like motif, mediates interactions with TPP1 [18,19]. POT1 and TPP1 act together and form a heterodimer now considered as the functional entity that protects the telomeric tail [18,19]. TPP1 interacts with POT1 thanks to a central motif called PBM (POT1-binding motif, Figure 2) [18]. Although bearing an OB-fold [20], TPP1 was not shown to bind DNA on its own; however, it stimulates POT1 binding on its target [21]. The heterodimer is connected to the rest of the Shelterin complex through interactions between TPP1 and TIN2 via a C-terminal TBM (TIN2-binding motif, Figure 2) [22]. These interactions allow the recruitment of TPP1 and POT1 to telomeres [23]. TPP1-POT1 plays essential roles in telomere homeostasis (Figure 1); it prevents the binding of RPA to the single-strand telomeric tail, avoiding activation of the ATR kinase and of the Alt-NHEJ (alternative NHEJ based on MRN/CtIP, PARP1 and ligase 3, amongst others [24]) pathways, which would fuse telomeres together otherwise [25]. More recently, POT1 was also shown to inhibit homology-directed DNA repair (HdDR) [26]. The outcome of POT1 loss was proposed to depend on the cellular model used, owing to differences in the redundancy and abundance of telomeric or repair proteins [26]. The heterodimer also controls the resection necessary to reform the overhang after DNA replication [27], recruits Telomerase through interactions between TPP1 and several domains of the enzyme [28,29,30,31] and regulates its activity with the help of TIN2 [32]. Several variants have been described for POT1 and have been shown to be associated with various types of cancer [33] and mutations in TPP1, particularly in the domains interacting with Telomerase, have been shown to cause a *Dyskeratosis congenita*-like phenotype, a telomeropathy which is characterized by abnormal shortening of telomeres and bone marrow failure [34]. TPP1 can also exist in two distinct isoforms, TPP1-S and TPP1-L (short and long, due to the different lengths of their N-terminal domains), with the former mainly being found in somatic cells, while the latter is more specific for differentiated male germ cells [35].

TRF1 and TRF2 bind and deal with the telomeric double-strand parts of telomeres. Structurally, these two proteins are closely related, since they originate from the same ancestral gene that was duplicated around 500 Myr ago at the basis of the vertebrate lineage [36]. Two domains have been evolutionary conserved (Figure 2): a central domain called TRFH (TRF Homology) [37], which structurally very similar between the two proteins (Figure 2); and a C-terminal Myb/SANT domain, called telobox, which provides sequence-specificity for the telomeric DNA tract [9,38,39]. The TRFH domain not only allows homodimerization of these proteins, but also constitutes an interaction hub for many partners [40,41]. Subtle variations in sequence between the two proteins provide specificity for different partners: TRF1 uses the TRFH to interact with TIN2, while in TRF2, the same region exhibits more affinity for various proteins involved in DNA repair (e.g., SLX4 [42], APOLLO [40,41], NBS1 [43] microcephalin [44]). This domain is also capable of binding DNA in a non-sequence-specific manner, and more precisely can wrap DNA around its circumference in a right-handed orientation [45]. This property was shown to allow TRF2 to modify DNA topology in vitro and in human cells [45,46]. Mutations of residues involved in this wrapping prevented TRF2 from forming a protective structure called the T-loop (Figure 4), whereby the telomeric single-stranded tail is buried in the double-strand and is protected from nucleolytic attack and illegitimate repair [45,47,48,49]. Recent results have suggested a different behavior in mouse cells (MEFs from *Mus musculus*), where telomeres are much longer (up to 50 kb versus the shorter human 4 to 10 kb) [50]. Formally, the TRFH domain of TRF1 is also capable of wrapping, although the presence of a very acidic sequence at the N-terminus of the protein (A domain, 76 residues, with an overall pHIof 3.5) prevents TRF1 from doing so [36]. Indeed, the N-termini of TRF1 and TRF2 are very divergent, both in terms of sequence and function. In place of the TRF1 acidic N-terminal domain, TRF2 bears a very basic sequence (B domain, Figure 2, pHI around 11.8) of either 45 or 87 residues, depending on the ATG codon used [15]. This domain is the third region of TRF2, which is capable of DNA binding, and more specifically recognizes branched structures in vitro [51,52]. Thanks to this binding, it protects Holliday junctions from resolution in vitro [52] and in cells [53,54]. This domain also binds G4 DNA structures [55] (DNA conformations formed by Hoogsteen base-paring of runs of 4 guanines [56]), interacts with the telomeric RNA TERRA [57] and promotes RNA invasion into DNA to create telomeric R-loops (a property inhibited by TRF1 [58]). The B domain of TRF2 also allows the interaction with several proteins, particularly DNA polymerase β, FEN1 and ORC1 [57,59,60]. Meanwhile, the TRF1 acidic domain allows interaction between TRF1 and a telomere-specific poly(ADP-ribose) polymerase, Tankyrase 1 [61], which is involved in telomere length regulation and resolution of sister telomere cohesion and which parylates TRF1 after telomeric DNA oxidative damage [62,63,64] in human cells. Of note, this interaction is absent in mouse cells, one of many differences in telomeres biology that have been reported throughout the years between these species. Finally, the domain falling between the TRFH and the Telobox (called the linker or hinge domain, Figure 2) is also very different between the two proteins; being much shorter in TRF1 (97 residues versus 190), it contains the binding site for the RecQ helicase BLM, which assists TRF1 in its replicative role (more below) [65], while in TRF2, it bears the binding sites for TIN2 and RAP1 of the Shelterin complex (Figure 2) and a short sequence that is involved in TRF2-mediated inhibition of the telomeric DNA damage response (iDDR) [66]. Overall, it is clear that despite their common origin, TRF1 and TRF2 have distinct partners and intrinsic properties, and this dichotomy is even more pronounced when considering their biological roles.

## 4. The Distinct Biological Roles of TRF1 and TRF2

TRF1 is mainly involved in replication and elongation of telomeric chromatin (Figure 3). Indeed, its removal causes activation of the ATR/CHK1 DDR pathway and telomeric replication defects that resemble those of fragile sites [13,67], hence the name of fragile telomeres given to this phenotype [13]. This fragility was recently shown to rely on break-induced replication (BIR) proteins, particularly POLD3 and POLD4 [68]. Interestingly, the fork stalling at the origin of the damage was proposed to be caused by G4 structures and the breakage was shown to involve SLX4 and its partner SLX1 [68]. Knowing that TRF2 binds both G4 and SLX4, it is not too far-fetched to think that TRF2 might be the culprit in TRF1/BLM-loss-mediated telomere fragility. Conversely, one may even propose that TRF1 acts to avoid TRF2 involvement, giving support to the argument for functional independence between the two proteins. TRF1-mediated replicative protection has recently been shown to be less efficient in hypoxic conditions in mouse cells, a deficiency corrected in the naked mole rat (the rodent with the longest lifespan, which can tolerate up to 15 min of total anoxia) through sequence adaptation in the TRF1 TRFH domain [69]. Besides telomeric replication, TRF1 is also involved in mitosis in relation to the mitotic spindle [70,71], meiosis to anchor telomeres to the nuclear envelope [72] and is important for the induction and maintenance of the pluripotent state in mouse embryonic stem cells through an indirect mechanism involving the telomeric RNA TERRA and the repressive PRC2 complex [73]. Conversely, TRF2 is dispensable for telomere protection in the same cells [74,75], showing again the differences in role and behavior between TRF1 and TRF2. As a whole, even if TRF1 in effect protects telomeres against damage, it is more dedicated to replicating the internal telomeric tract than actually capping the telomere terminus.

TRF2, on the other hand, is the major protector of telomeric ends (with TPP1/POT1 through different mechanisms; see above). Specifically, it inhibits DDR, homologous recombination and the NHEJ pathways; hence, phenotypes caused by its removal differ from the one observed after TRF1 loss and include ATM/CHK2-dependent DDR activation and telomeric fusions due to classical NHEJ (ligase-4-dependent pathway) [76,77,78,79,80,81,82]. The mechanisms behind this protection involve: the folding of telomeric DNA into T-loops [47,50,83]; the recruitment of RAP1 as a failsafe mechanism if T-loops cannot be formed, such as when telomeres are too short [45,84]; the sequestration of an unphosphorylated form of NBS1, which prevents its association with MRE11-RAD50 and the activation of ATM-dependent repair [43]; the inhibition of CHK2 activation by direct binding [76]; interacting with Ku70 and inhibition of its action [85]; suppression of RNF168 activation thanks to the iDDR motif [66]; limitation of HR thanks to the basic N-terminal domain (see above) [52,53,54,82]; protection of Holliday junctions against resolution [52,53,54] and dissolution by the Werner syndrome helicase [86]; blockage of telomere sister–chromatid exchange [87]. None of these pathways involve TRF1; however, TRF1 and TRF2 have one thing in common—they both facilitate telomeric replication. Even so, the mechanisms behind this and the proteins involved are different (inhibition of BIR and recruitment of BLM for TRF1 [65,68], recruitment of RTEL1 [88,89] and APOLLO [14,90,91,92] for TRF2; Figure 3). Because of the epistatic relationship between TRF1 and BLM, which has been shown to resolve G4 structures [93], it has been proposed that TRF1 would recognize forks blocked by the formation of these conformations on the lagging strand [13,68] and recruit BLM, although what exactly TRF1 recognizes is still unknown. The situation seems clearer for TRF2. TRF2 is able to recognize G4 structures [55] and Holliday junctions, as well as reverse forks [51,52], positively supercoiled DNA [14] and T-loops [49], all of which would stop fork progression. TRF2 will then recruit the appropriate activities required to alleviate the problem. Indeed, TRF2 interacts with many DNA processing enzymes, including FEN1 [59]; SLX1, ERCC1-XPF and MUS81-EME1 through SLX4 [42]; RTEL1 [89,94]; WRN [95]; and APOLLO [90,91]. Apart from T-loops which are telomeric by definition, all other structural elements recognized by TRF2 can be present during replication all over the genome; thus, it is not surprising that TRF2 is able to act outside telomeres. Indeed, it also exerts a protective role in the difficult-to-replicate pericentromeres, the larger constitutive heterochromatic region of the human genome, where it helps fork progression by recruiting RTEL1 [11,88].

Besides replication elongation, TRF2 also influences the initiation step. Indeed, it regulates origin activation on telomeres and pericentromeres by binding and recruiting ORC2 [96,97,98,99,100]. Finally, and perhaps more surprising, is the transcriptional role of TRF2. Indeed, TRF2 controls the transcription of several genes involved in mitochondrial metabolism, innate immunity, oncogenesis and neuronal development [101,102,103,104,105], linking telomeres to several cellular pathways determining cell fate.

## 5. RAP1: An Accessory Protein for Telomere Protection

RAP1 shares several properties with its budding yeast ortholog ScRap1, combining telomere and extra-telomere functions. As with ScRap1, it regulates metabolic genes [106,107] and protects telomeres against NHEJ [84]. This latter property was originally described in vitro [108,109,110] and remained controversial for several years. Indeed, RAP1 loss did not lead to the expected DDR activation and telomere fusion in mouse and human cells, and KO mice had metabolic and inflammatory, not telomeric, phenotypes [107,111,112,113,114,115]. This apparent discrepancy was later explained by the discovery that RAP1 anti-NHEJ function was a backup pathway in the event of TRF2 being unable to form T-loops or when telomeres were too short for TRF2 to do so, as in senescent cells or in Telomerase-negative mice [45,84,116]. Accordingly, RAP1 does not act as an anti-NHEJ factor in MEFs where telomeres are long [50], another example of the differences between *Mus musculus* and humans when telomere biology is concerned. The mechanism behind RAP1-mediated protection was proposed to involve an interaction with Ku, which would prevent its tetramerization and block c-NHEJ [85]. RAP1 is recruited on telomeres by TRF2 thanks to interactions between the RCT domain of RAP1 (Figure 2) and a RAP1-binding motif (RBM) in the hinge or linker domain of TRF2 [117,118]. A weaker interaction has also been described between the N-terminal sequence of RAP1 and the TRFH of TRF2 [119]. Falling into the same affinity range as the interaction between TRF2 and TIN2 in the same position, although much lower than that with APOLLO (or SLX4), this secondary binding could prevent the inappropriate binding of TIN2 on the TRFH, although would not be sufficient to displace APOLLO or SLX4 [119]. Interestingly, the transcriptional activities of RAP1 seem to be TRF2-independent. Finally, a cytoplasmic form of RAP1 is also involved in controlling inflammation, since it regulates NFkB activation [115].

## 6. TIN2: The Bridge

The last member of the complex, TIN2, has the capacity to bind TRF1, TRF2 and TPP1 (Figure 1). The main interaction binding sites of TRF1 and TRF2 lie on different surfaces of the TIN2 protein (Figure 2), one located in the N-terminal domain (for TRF2; this domain also contains the binding site for TPP1) and one located centrally (TBM, for TRF1) [22,40]. It is, therefore, involved in linking the double-stranded DNA binders TRF1 and TRF2 to the TPP1-POT1 heterodimer that deals with the telomeric single-stranded overhang. Three isoforms of TIN2 have been identified, named TIN2S, TIN2L and TIN2M [32,120] according to the length of their C-terminus. They all seem to participate in telomeric functions, although the extension found in TIN2L was shown to increase TIN2 interactions with TRF2 [121] and to cause the binding of TIN2 to the nuclear matrix [120]. TIN2 is crucial for the formation of the complex, of course by bridging TRF1 or TRF2/RAP1 to TPP1/POT1, but also by stabilizing TRF1 and TRF2 on telomeres [122,123,124]. It is also absolutely necessary in recruiting TPP1 and POT1 [22,23]. As with several of the telomeric proteins, TIN2 mutations are linked to human diseases, in this case *Dyskeratosis congenita* (DC). Mutations in TIN2 that lead to DC concentrate in a cluster, called the DC cluster, next to the TBM (Figure 2). How these mutations affect TIN2 behavior and cause the disease is still elusive; however, a molecular link has been observed between the most commonly mutated residue in DC, R282, and a serine located in the C-terminal extension in TIN2L (S396), although its role in the etiology of the disease is unknown [121]. The DC cluster also contains a binding site for heterochromatin protein 1γ, an interaction that is important for the establishment and maintenance of telomere cohesion during S phase [125]. Indeed, cell lines derived from patients exhibit defects in telomere cohesion [125]. TIN2 participates in Telomerase recruitment through TPP1 [126] and was shown to stimulate its activity in vitro [32], although it is not clear whether this property is related to DC. Indeed, the most common TIN2 mutations do not affect this activity, while the telomere shortening observed in a DC mouse model was Telomerase-independent [127]. Finally, TIN2 has also been seen to shuttle to the mitochondrion, where it controls mitochondria morphology and metabolism [112,128].

## 7. Shelterin Quaternary Structure

Leading teams in the field have described the molecular links between the six proteins, while the 3D structures of parts of the complex involving domains of several of these proteins have been published; hence, it is clear that connections between the mammalian Shelterin proteins do exist, with the notable exceptions of the mouse Pot1a and Pot1b proteins, for which no evidence has been reported of their co-existence in a telomeric complex. The presence on telomeres of the whole complex containing all Shelterin subunits and the real quaternary structure of this telomeric complex remain elusive. To ascertain the quaternary structure of the Shelterin complex, one would have to answer three questions: (i) Is it possible to form a whole Shelterin complex? (ii) Is there evidence that a whole complex physically exists on telomeres? (iii) Is there biological evidence of functions requiring the coordinated action of the 6 members?

(i) Forming the whole 6 proteins complex would require the absence of steric hindrance between subunits and an appropriate stoichiometry between the components. A complex containing all 6 proteins was repeatedly observed after chromatographic separation of nuclear proteins extracts [123,129,130,131]; thus, steric hindrance does not seem to exist, at least in the absence of DNA, since these experiments involve removal of nucleic acids. De Lange and collaborators [132], as well as Cech and colleagues, reconstituted a DNA binding complex in vitro containing the 6 proteins for the former and (TRF2-RAP1)_2_-TIN2-TPP1-POT1 [133] for the latter, supporting the argument against steric hindrance in the presence of DNA. The 5 members containing the complex obtained by Cech and colleagues might seem rather puzzling, although alternative binding of TIN2 on TRF2 in the TRFH domain (rather than in the hinge) was observed and studied in vitro by Ming and his collaborators [40]. It is, therefore, possible to imagine a complex containing two TRF2 dimers, both bound on TIN2, one through the TRFH domain and one through the hinge domain. Although weak, this alternative TRFH binding could be of importance in telomere functions. Indeed, mutations in the TIN2-binding motif abrogating both the interaction with TRF1 and this secondary binding with TRF2 do not cause TRF1 KO-like or TPP1-POT1 deletion phenotypes, but rather cause telomeric defects similar to those observed when deleting TRF2 (activation of ATM/CHK2, fusion by classical NHEJ) [22,134]. Regarding the stoichiometry, TRF1, TRF2 and TIN2 are roughly 10 times more abundant compared to TPP1-POT1 in the chromatin-bound fractions of several human cells [135]. This stoichiometry tells us that if a whole complex exists, it could not involve more than 10% of the TRF1, TRF2, RAP1 and TIN2 proteins. It also clearly implies the existence of subcomplexes, several of which have been observed using the chromatographic experiments cited above (Figure 4) [123,129,130,131]. Another piece of evidence for the existence of subcomplexes is the different dynamics measured by FRAP between TRF1, TRF2 and POT1—TRF1 has a residence time of ~44 s, which was also found for ~70% of TRF2 molecules; however, the POT1 residence time was much higher (~11 min) and was shared by the remaining ~30% of TRF2 molecules [136]. As such, TRF1 and TRF2 can be associated with subcomplexes of different dynamics, and importantly POT1 seems to behave differently than TRF1.

The compositions of these various subcomplexes are expected to vary from one cell type to another or to depend upon certain conditions (experimental or pathological) and upon the age of the cell. Indeed, telomeric proteins levels have been shown to change when cells or tissues age. TRF2 amounts were shown to decrease in human skeletal muscle over lifetime, while the other Shelterin proteins remained constant [102]. Similarly, TRF2 expression was downregulated upon senescence via a p53/Siah1-mediated pathway in normal human fibroblasts in culture [137,138], while in zebrafish a trend toward general downregulation of Shelterin gene expression was observed, with the exception of RAP1, for which mRNA expression decreased more rapidly in the intestines and the gills [139]. Another aspect related to telomere length, although in a pathological context, is that TRF2 and TPP1 (but not TRF1 and POT1) levels specifically decrease in aborted fetus material from idiopathic recurrent pregnancy losses, where telomeres are abnormally short [140]. We do not yet know the impacts of these variations in protein dosage on the nature of the telomeric subcomplexes, although the impressive downregulation that has been sometimes described (over 70% for TRF2 [102,140] and over 90% for TPP1 [140]) certainly argues for changes in the telomeric subcomplexes.

(ii) The presence of the whole complex on telomeres can perhaps be evidenced by studying the interdependence between the subunits. Both human and mouse POT1-TPP1 complexes are tethered to telomeres via TIN2 [22,23,141]. *In vitro*, TPP1-TIN2 has a greater affinity for TRF2 than TIN2 alone [142], and similarly in 293T cells, exogenously expressed Flag-TPP1 promotes the interaction between TIN2 and TRF2 (also exogenously expressed). This could be explained by the proximity of the binding sites of TPP1 and TRF2 on TIN2 (both bind on its TRFH domain), and suggests a hierarchical construction of the complex from TRF2 to POT1 [142,143]; however, here TRF1 is the odd one out. Indeed, TPP1 does not promote TIN2-TRF1 interaction [131] and removal of TRF1 from telomeres does not alter the telomeric content of TIN2-TPP1 [144]; however, depleting cells of TPP1 inhibits the capacity of RAP1 (and TRF2) to co-immunoprecipitate TRF1 [131]. Furthermore, in 293T cells, exogenously expressed V5-tagged RAP1 is able to co-immunoprecipitate all 5 other proteins (also exogenously expressed), unless TIN2 is absent, in which case V5-RAP1 precipitates only TRF2 [131]. Similarly, tagged TRF1-TIN2 was able to co-immunoprecipitate TRF2-RAP1 in HeLaS3 cells [123]. This, added to the fact that the 6 proteins containing the complex can be purified from HeLa cell extracts, argues for the existence of a complete Shelterin complex; however, the major caveats of these experiments are the absence of DNA and the extraction protocol that may alter or reorganize protein complexes. As such, we do not have a clear vision of the quaternary structure of the complexes as they are telomeres.

(iii) There is no clear evidence of telomeric functions that require the presence of the six members on the same complex. Indeed, several lines of evidence point to a functional independence between TRF1 and TRF2. TRF2 depletion causes a strong uncapping phenotype characterized by losses of the T-loop [48], ATM and activation and telomeric fusion of its downstream kinase CHK2 [78,79]. TRF1 clearly blocks the ATR-dependent DDR pathway and promotes telomeric replication [13,67], but does not affect T-loop formation [48]. Regarding telomere capping, conflicting results have been published on the ability of TRF1 to inhibit the activation of ATM/CHK2 and fusion [13,67]. Furthermore, telomere deprotection resulting from the expression of TIN2 mutants unable to bind TRF1 (L247E in mouse TIN2, L260E in human TIN2) [134] is rescued by increasing the cellular amount of TRF2 or by tethering TIN2 to TRF2 [134], suggesting that the attachment of TRF1 to TIN2 is not absolutely necessary for telomere capping. In further agreement with a functional independence between TRF1 and TRF2 in telomere capping, TRF1 and TRF2 have been shown in some instances to act in opposite ways—TRF2 causes formation of telomeric R-loops, a process inhibited by TRF1 [58], while conversely to TRF2 depletion, TRF1KD has been recently shown to protect short telomeres against recombination [145]. TRF1 is necessary for the establishment and maintenance of pluripotency in mouse ES cells, while TRF2 is dispensable [73,74,75]. Finally, an important feature of the telomere capping complex should be the recognition of the ds/ss junction of telomeric DNA. This has been shown to be a property of the TRF2-RAP1 complex [146], while a (TRF2-RAP1)_2_-TIN2-TPP1-POT1 complex has the stronger affinity for this DNA substrate in vitro [133].

## 8. Concluding Remarks

Overall, the above studies, added to the fact that mass spectrometry analysis of the telomeric complexes showed TRF1 interaction to be the weakest within Shelterin [147], raises doubts about the existence of a whole Shelterin complex being responsible for capping telomeric ends Only one instance, as far as we know, requires a coordinated action of TRF1 and TRF2, whereby TRF2 recruits the BUB1-BUB3 complex to telomeres during the S phase and BUB1 then phosphorylates TRF1 to stimulate the recruitment of BLM [148], although even then the presence of both proteins on the same complex has yet to be determined.

As an alternative model, one could hypothesize that telomeres are bound by several types of subcomplexes, some based on TRF1, others on TRF2, with different roles and locations (Figure 5). TRF2- but not TRF1-containing complexes could be in charge of T-loop-based end protection. These could be located at the far ends of telomeres, while TRF1-containing complexes might be more internal to comply with the more replicative role of TRF1. Other complexes most probably exist, such as ones centered on TRF2 in partnership with Apollo to rescue stalled forks [14] or containing only TRF1 and TIN2. Determining the nature and the roles of these subcomplexes will undoubtedly be a challenge but will be absolutely necessary to finally draw an accurate image of telomere organization.

## Figures and Tables

**Figure 1 cells-10-01753-f001:**
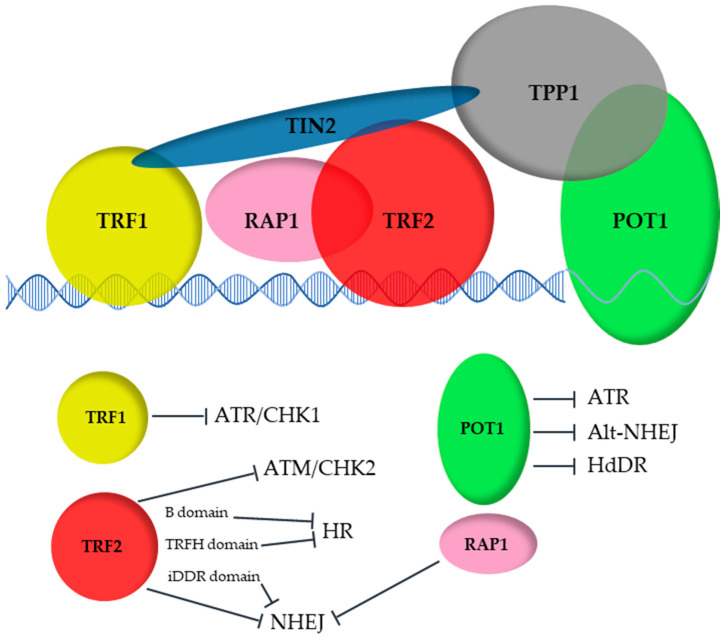
The Shelterin complex protects telomere from illegitimate activation of DNA damage response (ATM/CHK2, ATR/CHK1) and repair (homology-directed DNA repair, HdDR, NHEJ, HR).

**Figure 2 cells-10-01753-f002:**
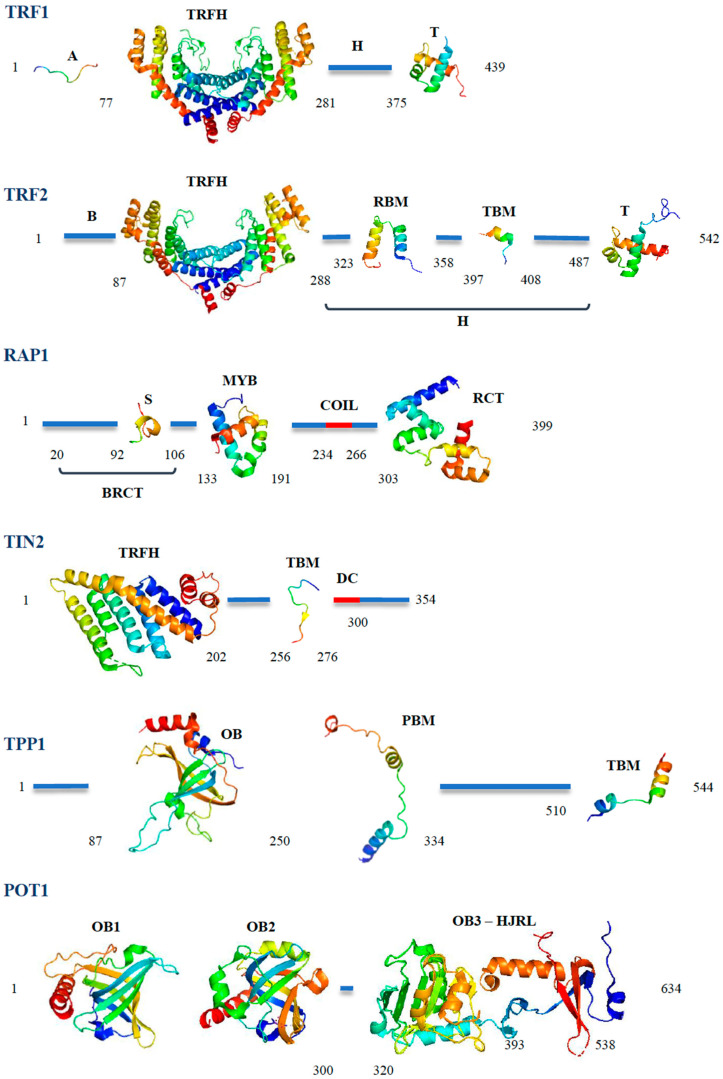
Structural elements of the Shelterin proteins. All structures were extracted from the PDB database using PyMol and are shown as rainbow-colored ribbons from blue (N-terminus) to red (C-terminus). TRF1: The four domains of TRF1 are labeled A (acidic domain, structure extracted from its complex with TNKS1, pdb 5hkp), TRFH (TRF homology domain, shown as a dimer, structure extracted from its complex with TIN2, pdb 3bqo), H (hinge domain) and T (telobox, pdb 1ity). TRF2: The four domains of TRF2 are labeled B (basic domain), TRFH (TRF homology domain, shown as a dimer, structure extracted from its complex with Apollo, pdb 3bua), H (hinge domain), RBM (RAP1-binding motif, structure extracted from its complex with RAP1, pdb 3k6g), TBM (TIN2-binding motif, structure extracted from the TRF2-TIN2-TPP1 complex, pdb 5xyf), iDDR (inhibition of the DNA Damage Response) and T (Telobox, pdb 1xg1). RAP1: The four domains of RAP1 are labeled BRCT (BRCA1 C terminal domain), which contains a secondary binding site for the TRF2 TRFH domain (S, structure extracted from its complex with TRF2 TRFH, pdb 4rqi), MYB (Myb domain, pdb 1fex), coil (coil domain) and RCT (RAP1 C terminal domain, structure extracted from its complex with TRF2, pdb 3k6g). TIN2: The three domains of TIN2 are labeled TRFH (TRF homology domain, structure extracted from the TRF2-TIN2-TPP1 complex, pdb 5xyf), TBM (TRFH-binding motif, structure extracted from its complex with TRF1, pdb 3bqo) and DC (*Dyskeratosis Congenita* hotspot). TPP1: The three domains of TIN2 are labeled OB (Oligosaccharide-Binding fold, pdb 2i46), PBM (POT1-binding motif, pdb 5h65), TBM (TIN2-binding motif, structure extracted from the TRF2-TIN2-TPP1 complex, pdb 5xyf). POT1: The four domains of POT1 are labeled OB1 and OB2 (oligosaccharide-binding fold 1 and 2, pdb 1xjv), OB3 and HJRL (oligosaccharide-binding fold 3 and Holliday Junction Resolvase-like domain, structures extracted from the complex with TPP1, pdb 5h65).

**Figure 3 cells-10-01753-f003:**
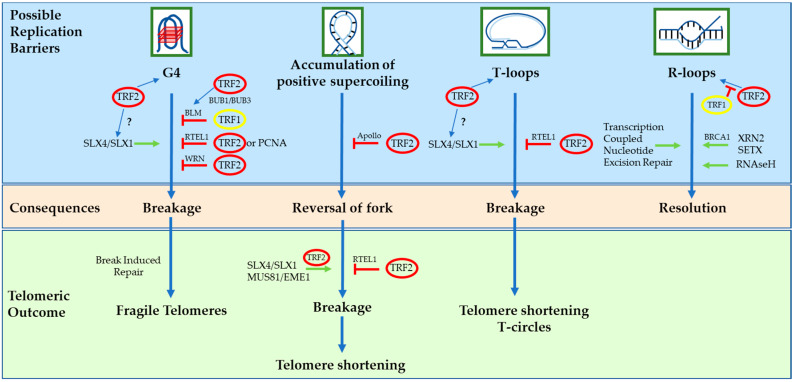
Schematic view of the many ways TRF1 and TRF2 can influence telomeric replication and avoid deleterious effects on telomere size or state.

**Figure 4 cells-10-01753-f004:**
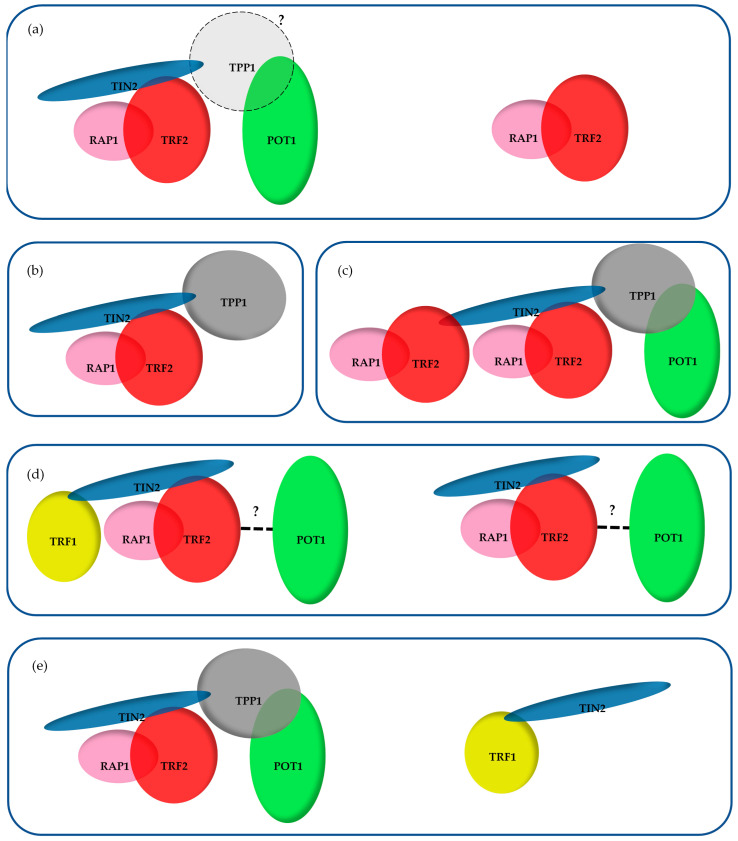
Subcomplexes observed following chromatographic separation of nuclear proteins: (**a**) Ye et al. [123]; (**b**) Giannone et al. [129]; (**c**) Lim et al. [133]; (**d**) O’Connor et al. [131]; (**e**) Kim et al. [130].

**Figure 5 cells-10-01753-f005:**
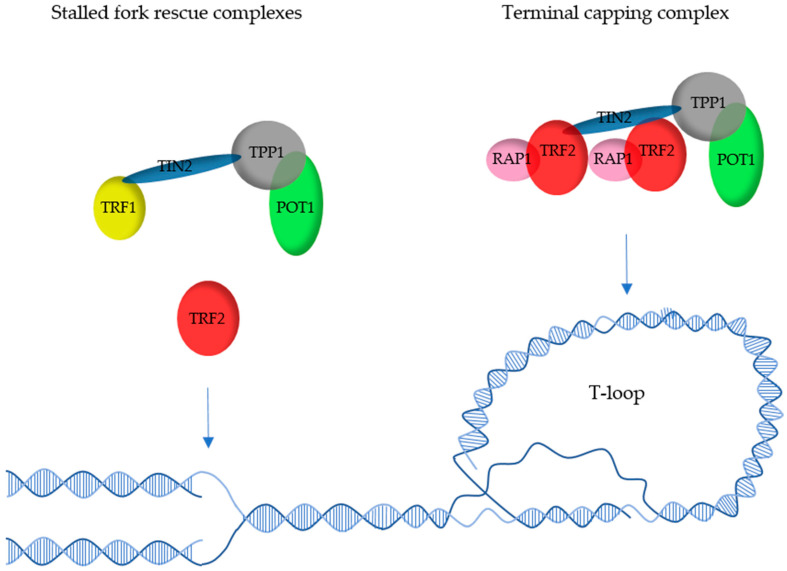
Possible complexes involved in telomere replication and T-loop-based capping. Fork rescue has been shown to involve TRF1 [13], most probably as a TRF2-independent complex [65], and TRF2 independently of TRF1 [14,92]. On the other hand, T-loop-dependent capping involves TRF2 [50], and the single-strand overhang dynamics are regulated by TPP1-POT1 [25], most probably as a TRF1-independent complex.

## Data Availability

No new data were created or analyzed in this study. Data sharing is not applicable to this article.

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
