# Peer review of "Multifunctionality of the Telomere-Capping Shelterin Complex Explained by Variations in Its Protein Composition"

_cells, 2021, doi:10.3390/cells10071753_

Round 1

Reviewer 1 Report

This manuscript was excellently written, which describes the 6 shelterin proteins along with their functions alone or with their interacting proteins. Most interestingly, the authors discussed the possibility that the multifunctionality of the shelterin complex is determined by the formation of different subcomplexes. All the figures were well composed, and all the references were appropriately cited. I have minor comments:

  1. Figure 3 and 4. The authors may consider to change the order of these figures.
  2. Page 8.”Because of the epistatic relationship between TRF1 and BLM, which has been shown to resolve G4 structures, it has been proposed that TRF1 would recognize forks blocked by the formation of these conformations on the leading strand”. Is the formation of G4 on the lagging strand?
  3. It would be informative, if the authors would consider adding a new figure showing how TRF1 and TRF2 regulate telomere replication.

Author Response

Journal Cells

Manuscript ID cells-1258025

Multifunctionality of the telomere capping shelterin complex explained by variations in its protein composition

Marie-Josèphe Giraud-Panis * , Eric GILSON * , Claire Guilain

Many thanks for your helpful comments. We have taken your remarks into account and have changed the text and figures accordingly. Hopefully, these changes will meet your approval.

Best Regards

The authors

  1. Figure 3 and 4. The authors may consider to change the order of these figures.

Done

  1. Page 8.”Because of the epistatic relationship between TRF1 and BLM, which has been shown to resolve G4 structures, it has been proposed that TRF1 would recognize forks blocked by the formation of these conformations on the leading strand”. Is the formation of G4 on the lagging strand?

Indeed, changed in the text.

  1. It would be informative, if the authors would consider adding a new figure showing how TRF1 and TRF2 regulate telomere replication.

We have added a new Figure 3 showing the different ways TRF1 and TRF2 can facilitate replication.

Reviewer 2 Report

The review by Ghilain et al aims to review our knowledge of shelterin subcomplexes and the functional independence between them. The content is detailed and well researched.

Overall, I think the authors fall short on their aim. The first half of the review is a general overview of each individual shelterin sub-component. This background section is too long with certain points being repeated in more detail in the second half of the review.

The flow of information is also disjointed in places e.g. (page 1 paragraph 1)  " Indeed, several premature aging syndromes generically called telomeropathies originate from or are associated with mutations in telomere associated proteins  (Dyskeratosis congenita, Hoyeraal-Hreidarsson, Revesz or Coats Plus syndromes amongst others)[4] and activating pathways that replenish telomeres (Telomerase or Alternative Lengthening of Telomeres, ALT, based on recombination) is an obligatory step for oncogenesis."

"Telomerase, which is composed of a catalytic subunit reverse transcriptase (TERT) and an associated RNA template (TERC), can compensate for this inexorable replicative erosion that is caused by Telomerase down-regulation in human somatic cells."

This section should be either cut or shortened significantly.

The review becomes more interesting from page 7 onwards, with the paragraphs detailing the differences between TRF1 and TRF2 function being informative and of interest to readers. However, these sections do not address the main aims of the review outlined in the abstract.

The last 1/4 of the review addressed the subject matter and provides a solid case for different subcomplexes of shelterin existing in the cell. This section should be substantially expanded upon. Some topics to address could include:
1) What do the authors think the different shelterin subcomplexes are doing?
2) What do the authors think about the lack of steric hindrance in the 5-member shelterin subcomplex described by Tom Cech's group?
3) Expand upon the changes in shelterin expression during aging.

Author Response

Journal Cells

Manuscript ID cells-1258025

Multifunctionality of the telomere capping shelterin complex explained by variations in its protein composition

Marie-Josèphe Giraud-Panis * , Eric GILSON * , Claire Guilain

Many thanks for your helpful comments. We have taken your remarks into account and have changed the text and figures accordingly. Hopefully, these changes will meet your approval.

Best Regards

The authors

The background section is too long with certain points being repeated in more detail in the second half of the review.

We feel that giving background knowledge of the different roles of these proteins is not only important for the reader but is also stressing the functional independence between some of the Shelterin members (namely TRF1 and TRF2).

The flow of information is also disjointed in places.

The sentences cited by this reviewer have been modified in page 1:

In somatic cells, this leads to an inexorable erosion of chromosomes ends which is compensated by activating pathways that replenish telomeres (Telomerase or Alternative Lengthening of Telomeres, ALT, based on recombination) in germ, stem and cancer cells [2].

1) What do the authors think the different Shelterin subcomplexes are doing?

We have modified the text in page 12.

As an alternative model, one could hypothesize that telomeres are bound by several types of sub-complexes some based on TRF1, others on TRF2 with different roles and location (Figure 5). TRF2 but not TRF1 containing complexes could be in charge of T-loop based end protection and thus be located at the far end of telomeres, while TRF1 containing complexes might be more internal to comply with the more replicative role of TRF1. Other complexes most probably exist such as ones centered on TRF2 in partnership with Apollo to rescue stalled forks[14] or containing only TRF1 and TIN2. Determining the nature and the role of these subcomplexes will undoubtedly be a challenge but will be absolutely necessary to finally draw an accurate image of telomeres organization.

2) What do the authors think about the lack of steric hindrance in the 5-member shelterin subcomplex described by Tom Cech's group?

We have added the following paragraph in page 9 of the review:

The 5 members containing complex obtained by Cech and colleagues might seem rather puzzling but an alternative binding of TIN2 on TRF2 in the TRFH domain (rather than in the Hinge) has been observed and studied in vitro by Ming Lei and his collaborators[39]. It is therefore possible to imagine a complex containing two TRF2 dimers, both bound on TIN2, one through the TRFH domain and one through the Hinge domain. Although weak, this alternative TRFH binding could be of importance in telomere functions. Indeed, mutations in the TIN2 Binding Motif abrogating both the interaction with TRF1 and this secondary binding with TRF2 do not cause a TRF1 KO-like or TPP1-POT1 deletion phenotypes but rather cause telomeric defects similar to those observed when deleting TRF2 (activation of ATM/CHK2, fusions by classical NHEJ)[21,133].

3) Expand upon the changes in shelterin expression during aging.

The following paragraph has been added in page 11:

The composition of these various sub-complexes is expected to vary from one cell type to another or to depend upon conditions (experimental or pathological) and upon the age of the cell. Indeed, telomeric proteins levels have been shown to change when cell or tissue age. TRF2 amounts were shown to decrease in human skeletal muscle over lifetime while the other Shelterin proteins remained constant[102]. Similarly, TRF2 expression is down-regulated upon senescence via a p53/Siah1-mediated pathway in normal human fibroblasts in culture [136,137] and in zebrafish a trend toward general down-regulation of Shelterin gene expression was observed, with the exception of RAP1 for which mRNA expression was decreasing more rapidly in the intestine and the gill[138]. Also related to telomere length but in a pathological context, TRF2 and TPP1 (but not TRF1 and POT1) levels specifically decrease in aborted fetus material from idiopathic recurrent pregnancy losses, where telomeres are abnormally short[139]. We do not know as yet the impact of these variations in protein dosage on the nature of the telomeric sub-complexes but the impressive down-regulations that have been sometimes described (over 70% for TRF2[102,139] and over 90% for TPP1[139]) certainly argues for changes in the telomeric sub-complexes.

Reviewer 3 Report

This is an excellent review about our current understanding of the shelterin complex by Ghilain et al. The only addition I would like to suggest is a graphical overview of the interaction of shelterin components with DNA damage to help the reader visualize the different interaction possibilities (e.g. NHEJ, Ku,...)

Author Response

Journal Cells

Manuscript ID cells-1258025

Multifunctionality of the telomere capping shelterin complex explained by variations in its protein composition

Marie-Josèphe Giraud-Panis * , Eric GILSON * , Claire Guilain

Many thanks for your helpful comments. We have taken your remarks into account and have changed the text and figures accordingly. Hopefully, these changes will meet your approval.

Best Regards

The authors

The only addition I would like to suggest is a graphical overview of the interaction of shelterin components with DNA damage to help the reader visualize the different interaction possibilities (e.g. NHEJ, Ku,...)

Figure 1 has been modified to schematize these functional interactions.

Reviewer 4 Report

In this review, Ghilain et al described a multifonctionality of the telomere capping shelterin complex. The manuscript covers several aspects of the subject. The paper is an update on the topic.  

However, the organization of the paper needs some modifications. The lack of clearly parts in the article makes it more difficult to follow.  

In the conclusion, the authors can underline the importance of these shelterin proteins in the clinical management of patients with telomeropathies as well as in the development of new therapies. 

Author Response

Journal Cells

Manuscript ID cells-1258025

Multifunctionality of the telomere capping shelterin complex explained by variations in its protein composition

Marie-Josèphe Giraud-Panis * , Eric GILSON * , Claire Guilain

Many thanks for your helpful comments. We have taken your remarks into account and have changed the text and figures accordingly. Hopefully, these changes will meet your approval.

Best Regards

The authors

However, the organization of the paper needs some modifications. The lack of clearly parts in the article makes it more difficult to follow.  

We agree with this reviewer and have added titled subdivisions.

In the conclusion, the authors can underline the importance of these shelterin proteins in the clinical management of patients with telomeropathies as well as in the development of new therapies. 

We have added the following sentences in page 1.

Thus, the obvious importance of telomeric actors has elicited various therapeutic trials. Although Telomerase could be perceived as a prime target, limitations in Telomerase-based strategies, such as the consequential activation of ALT-driven telomeres elongation or the possible pro-aging side effects, have stimulated the development of alternative approaches[6]. Targeting the complexes that form telomeres themselves could thus be a complementary or alternative route for telomere-based therapies.

Round 2

Reviewer 2 Report

The authors have mostly satisfied my concerns. However, I still think the first half of the review should be condensed and not overly repeated in the second half.